# Bibliometric Analysis of Artificial Intelligence in Textiles

**DOI:** 10.3390/ma15082910

**Published:** 2022-04-15

**Authors:** Habiba Halepoto, Tao Gong, Saleha Noor, Hafeezullah Memon

**Affiliations:** 1Engineering Research Center of Digitized Textile and Fashion Technology, Donghua University, Shanghai 201620, China; 317111@mail.dhu.edu.cn; 2College of Information Science and Technology, Donghua University, Shanghai 201620, China; 3School of Information Science and Engineering, East China Science and Technology University, Shanghai 200237, China; saleha.noor@yahoo.com; 4College of Textile Science and Engineering, Zhejiang Sci-Tech University, Hangzhou 310018, China; hm@zstu.edu.cn

**Keywords:** bibliometric analysis, textiles, research trend, artificial intelligence, Web of Science

## Abstract

Generally, comprehensive documents are needed to provide the research community with relevant details of any research direction. This study conducted the first descriptive bibliometric analysis to examine the most influential journals, institutions, and countries in the field of artificial intelligence in textiles. Furthermore, bibliometric mapping analysis was also used to examine diverse research topics of artificial intelligence in textiles. VOSviewer was used to process 996 articles retrieved from Web of Science—Core Collection from 2007 to 2020. The results show that China and the United States have the largest number of publications, while Donghua University and Jiangnan University have the highest output. These three themes have also appeared in textile artificial intelligence publications and played a significant role in the textile structure, textile inspection, and textile clothing production. The authors believe that this research will unfold new research domains for researchers in computer science, electronics, material science, imaging science, and optics and will benefit academic and industrial circles.

## 1. Introduction

Artificial intelligence has changed people’s lives, facilitating the performance of repeated tasks with maximized accuracy. Like in everyday life, artificial intelligence also finds its application in the field of textiles. Textile materials are characterized by flexibility, fitness, and fineness, and generally, they find their application in apparel or upholstery and, to some extent, reinforcements of textile composites [1,2]. Textiles are the second basic need of people after food, which makes their study worthwhile, as humankind’s daily life is connected to them [3]. The use of textile material dates to the stone age; it is used for shelter and has been a source of identity, showing one’s social status, gender, or culture [4]. Due to the rapid development in computer science in the last decades, there has also been much advancement in manufacturing, testing, and analyzing textiles [5]. Artificial intelligence finds its application from fiber development to fiber assembly in slivers, yarns, fabrics, or garments [6].

There are two ways to summarize research publications, i.e., review papers and bibliometric research analysis [7]. The bibliometric analysis may be used as a predictive measurement tool for experimental study and choosing the research direction for new coming researchers. The statistics derived from the bibliometric analysis quantify the contribution of scientific articles to a particular subject. They reflect current scientific developments and may be used to identify potential developments; thus, the next science pattern may be forecast by bibliometric analysis. Less often, the bibliometric analysis has been published in the field of textiles and garments. Tian and Jun have recently published a bibliometric analysis on protective clothing research [8]; Yan and Xu analyzed the textile patentometrics [9]; Feng and coworkers studied textile and clothing footprint [10]; however, their research tool was CiteSpace. One more interesting bibliometric analysis published in textiles is related to textile schools [11]. In previous literature, diverse disciplines have used VOSviewer to conduct bibliometric analysis, sustainable supply-chain management [12], sustainable design for users [13], international entrepreneurship [14], plant-based dyes [15], health promotion using Twitter [16,17], industrial marketing management [18], applied mathematical modeling [19], circular economy [20], exchange rate and volatility [21], and so on. Thus, there are very few studies related to textiles, and according to our best knowledge, none of the authors have studied artificial intelligence in Textiles using the VOSviewer software tool. This study was aimed to explore the following key questions.

What is the annual growth of publications in the field of artificial intelligence in textiles? What are their citation trends and usage counts in the database of Web of Science?How are the publications related to artificial intelligence in textiles distributed? What are the most influential countries, journals, and institutes?Which research group, country, and organization are most productive based on citations and bibliographies?What are the emerging topics related to artificial intelligence in textiles?How is the existing publication spread? What keywords are related to each other?

Through this bibliometric analysis research, this paper profoundly analyses the topic, i.e., artificial intelligence in textiles and its publication and citation worldwide, usage count and citation time analysis, the choice of authors, cooperation relationship between subjects, co-occurrence analysis of the words used in the abstracts, and cluster analysis of the manuscripts in this field. We believe that this research would help new researchers wisely select the research domain and provide a basic understanding of artificial intelligence’s current status in textiles.

## 2. Data Collection and Research Methodology

### 2.1. Data Source

Web of Science is considered the most reliable scientific and technical literature indexing platform capable of introducing the most important scientific and technological research fields. The data were retrieved on 31 January 2021, from Science Citation Index (SCI) Core collections, using search query in Appendix A. A total of 996 research papers related to textile image processing were published between 2000 and 2020. The citation counts for top-cited manuscripts were exported based on the SCI citation search method since it guarantees that the citing literature has gone through the scientific evaluation process before publication. The manuscripts with 100 citations were considered as top-cited manuscripts in this research. The journals’ impact factor is in accordance with the Journal Citation Reports published in 2019 since it is the most recent available data.

### 2.2. Bibliometric Methods

Here, we used VOSviewer to develop the mapping of the dataset. Since VOSviewer is a free software tool for constructing and visualizing bibliometric networks, these networks can include journals, researchers, or personal publications, which can be constructed based on citation, bibliographic coupling, co-citation, or co-author relationships. VOSviewer also provides a text mining function, which can be used to construct and visualize the co-occurrence network of essential terms extracted from a large number of scientific pieces of literature.

### 2.3. Inclusion and Exclusion Criteria

We started as a query string for topics related to artificial intelligence (i.e., image processing, image recognition, pattern recognition, machine learning, deep neural network, object recognition, and computer vision) and textiles (yarn, weave, knitted fabrics, hosiery, woven fabrics, drape, drapability, garments, and nonwovens) at the WoS Core Collection database. A total of 1145 results appeared in the timespan of 2007 to 2020. The publications comprised 11 languages: English (1127), Chinese (4), Spanish (3), Turkish (3), German (2), Japanese (1), Russian (1), Portuguese (1), Slovenian (1), and Croatian (1). As there was a significant difference among languages, we refined data to manuscripts written in English only. The remaining 1129 manuscripts comprised articles (724), early access (26), proceedings papers (398), reviews (21), data papers (2), editorial materials (3), and a correction (1), and thus, data paper, correction, and editorial material were excluded. Finally, 1123 papers were manually checked for their relevance to the topic. It was found that there were some overlapping terms, and thus, these articles were manually excluded from the dataset. For example, drape [22], weave [23], weaves [24], weaving [25,26], woven [27,28], weave ethics [29], weave the advantages [30], systems weave computing and communication [31], piece of music by weaving [32], “Weaving, Swerving, Sideslipping” [33], weaving sensible plots [34], WEAVE and 4MOST spectrographs [35], drape full-motion video, traffic-weaving [36], yarn [37,38,39], Hadoop yarn [40], yarn cluster [41], social fabric [42], and pressure-sensitive textiles [43]. It should be noted that the contextual meaning of these terms was not the same as the research domain of this manuscript. Thus, in total, 996 manuscripts were refined, and their distribution concerning the Web of Science Index was as follows: Science Citation Index Expanded (637), Social Sciences Citation Index (28), Conference Proceedings Citation Index-Social Sciences and Humanities (13), Conference Proceedings Citation Index-Science (339), Emerging Sources Citation Index (26), and Arts and Humanities Citation Index (5). The dataset of a final chosen manuscript might be requested from the corresponding author of this manuscript; its summary according to document type is presented in Table 1.

### 2.4. Data Analysis

The final dataset was exported from WoS and was analyzed in detail. This bibliometric research analyzed in-depth articles, topics, partnerships, times cited, co-words, and cluster analysis of papers. The VOSviewer was used to examine the co-occurrence mapping.

## 3. Results and Discussion

### 3.1. Global Publications and Citation Output

The publication output has been summarized in Figure 1a. It can be seen that the global publication of manuscripts in the field of artificial intelligence has risen. This rise in the number of publications has been divided into three time periods, i.e., 2007 to 2011 (old papers), 2012 to 2015 (recent papers), and 2016 to 2020 (current papers). It was observed that from 2007 to 2011, there was a sudden rise in publications, which became stable from 2012 to 2015, and then, currently, from 2016 to 2020, it has been increased dramatically. However, there was a continuous exponential increase in the number of citations, from 1 citation in 2007 to 1815 citations in 2020, as shown in Figure 1b (only citations from WoS Core collections were considered here). This research domain has shown to be promising, with an h-index of 36 and 7.37 citations per item. The total sum of citations is 7239 up to December 2020.

The usage count of scientific papers is directly related to the preference of readers. In general, readers from the scientific community prefer reading the latest article. However, highly cited manuscripts are often being used for a long time after their publication. This is what can be observed in Figure 2. The old manuscripts (2007–2011) possessed a higher number of citations, whereas they recently possessed a lesser usage count.

On the other hand, the current papers (2016–2020) have a higher usage count but lesser citations. The recent papers (2012–2015) had a higher number of citations and a higher usage count; they are at the maturity stage. This analysis agreed with the recent analysis related to the usage count versus citation from the Web of Science [44]. Herewith, citations of more than 100 in the Web of Science of all databases, highly cited manuscripts, are summarized in Table 2. We have listed the details, time cited, and usage count here.

### 3.2. Distribution of Publications

In total, 61 countries were contributing to the field of artificial intelligence in textiles. It would not be justifiable to compare publications by country; however, this is highlighted here to show the research domain’s distribution. According to their participation, the top countries are illustrated in Figure 3. The number of publications in these countries was as follows: China (321), USA (96), Iran (77), Turkey (68), India (67), France (57), Germany (54), Canada (36), Italy (31), Spain (31), and other (158). It was found that China has been the top participant in this research domain, followed by the USA.

In total, 944 organizations or institutes were found to be participating in this research domain. The most influential institutions are summarized in Table 3. Donghua University was considered the top publishing institute in this research domain, followed by Jiangnan University.

The most Influential Journals are summarized in Table 4. It can be seen that the *Journal of the Textile Institute* has remained the top choice for the authors for the given field to publish their work, which is followed by the textile research journal.

### 3.3. Subject Categories of Research Productivity

Based on the classification of subject categories in Web of Science, the publication output data of research related to artificial intelligence in textiles was distributed in 45 subject categories during the last fourteen years. Subject categories containing at least ten articles are shown in Figure 4. Three research fields were prominent for the given research direction, including material science, engineering, and computer science.

The co-occurrence map according to the Web of Science categories was plotted. The circle’s size describes the keyword’s potential; as shown in Figure 5, there are three clusters: Cluster 1 (Red) is mainly about the subjects related to computer science and artificial intelligence. The related subjects include cybernetic, hardware and architecture, imaging science and photographic technology, information system, information systems, software engineering, telecommunications, theory, and method. The second cluster (Green) is mainly about multidisciplinary fields covering physics, analytical and physical chemistry, nanoscience, and nanotechnology as a multidisciplinary science. The third cluster is related to material science, particularly textiles, polymer science, composite, and mechanics.

It should be noted that data mining and machine learning are also considered significant artificial intelligence fields; image processing uses both fields’ technology. Image processing’s research direction combines image processing and main methods and their application to intelligent detection, recognition, and classification. First, a unique technique is used to capture the required image data, and then, the image data are processed [52]. Finally, computer vision and related digital image tools are used for analysis [53].

### 3.4. Co-Occurrence of Keywords in the Abstracts

Here, we identified the keywords from the abstracts of all the manuscripts related to artificial intelligence and textiles from the final dataset. The circle’s size describes the keyword’s potential, as presented in Figure 6, while the line’s thickness was kept constant regardless of the link strength. The co-occurrence map based on the text data was plotted using the VOSViewer under the binary counting method with a ten-or-more threshold frequency. The network visualization graph under the association method with weight as the occurrence plotted the 18,883 terms; 403 met the threshold, and the 60% most relevant terms were plotted. It was found that there were three clusters (even with the cluster size = 1). Interestingly, each cluster presented a unique research theme in the field of textile engineering.

#### 3.4.1. Cluster 1 (Red): Artificial Intelligence for Textile Structures

The textile structure might be understood as the fiber structure; see the main keywords, i.e., ratio, density, diameter, and morphology. It may also be for yarn structure; see the main keywords, i.e., yarn hairiness, yarn count, yarn hairiness, yarn diameter, yarn property, yarn image, and twist. Moreover, the fabric structure may be related to keywords of fabric surface, wrinkle, hole weft, warp, porosity fabric appearance, fabric sample, stiffness, weft direction, surface roughness, and weft yarn. The textile structure cluster also covers the nonwoven and composite mechanical, in which nonwoven is mainly linked to fiber. However, the composite is linked with property, failure, geometry, layer image processing technique, image processing method, image processing software, image processing algorithm, and digital image processing technique are standard terms used to process the data. However, in particular, yarn is linked with the technique of artificial neural networks, i.e., Kalman filtering [54] or yarn color [55]. Property, effect, factor, behavior, distribution, change, test, and influence are common keywords related to every subgroup in the cluster.

#### 3.4.2. Cluster 2 (Green): Artificial Intelligence for Textile Inspection

The fabric defect detection in artificial intelligence in textiles has remained of particular interest for researchers [56,57]. This cluster mainly focuses on the defect, recognition, classification, and inspection in the textile industry that promise quality control. The keywords are primarily related to woven fabric and fabric defect, in which computer vision is used for fabric defect and fabric texture (weave pattern) detection using several approaches such as pattern recognition, design methodology approach, support vector machine, neural network, convolutional neural network, feature extraction, deep learning, image segmentation, and the genetic algorithm of image processing technology. Visual inspection is replaced with automatic detection by training the classifier for color and texture using fabric images [58]. Automation (machine vision) has been proposed for the segmentation of databases into class using the filter to better results without complexity as a novel method to overcome the traditional method.

#### 3.4.3. Cluster 3 (Blue): Artificial Intelligence for Textile and Apparel Production

This cluster focuses on the latest garment technology, particularly sensors for the product and production, to overcome the challenge and tasks. The manuscripts in this cluster mainly discuss the concept of monitoring the environment or activity of patients or persons in real-time. Moreover, it proposes smart textiles for the robot to replace hardware in IoT at low cost as a prototype for the designer in the apparel, clothing, and fashion industry. This cluster discusses data mining, image acquisition, and machine learning algorithms to meet consumer demand for textile products in the future.

## 4. Conclusions

This study summarizes a vital research domain, artificial intelligence in textiles, in one glance. This study suggests that this research field has remained a good discipline over the last 14 years and shall remain of particular interest in coming years. Thematic analysis revealed that artificial intelligence in textiles is not limited to pattern recognition. The analysis covers all major fields of artificial intelligence, including data mining and machine learning. Some key findings of this research include that the most dominant country was China, with 321 total publications. The USA was the runner-up with 96 total publications. Donghua University and Jiangnan University have the highest number of publications. Kumar [45] from the Indian Institute of Technology, got 312 citations in WoS core collections, while Ngan et al. [46] University of Hong Kong got 138 citations in WoS core collections. Co-occurrence of keywords in the abstracts yielded three major themes, i.e., artificial intelligence for textile structures, textile inspection, textile, and apparel production.

It should be noted that despite the extensive analysis, this research has some limitations. For instance, this analysis is based on the data provided by WoS, which is, of course, one of the authentic and accurate sources of information; however, the trend might be different when adding some other search engines or databases as well as when including manuscripts other than core collections. This study did not cover reviewer’s information, such as which institution or reviewers reviewed the manuscripts in this given area. Therefore, the research in this field needs to be further deepened. Moreover, the in-between linkage choice of the journal, editor, and selected reviewer might also be recommended for future research.

## Figures and Tables

**Figure 1 materials-15-02910-f001:**
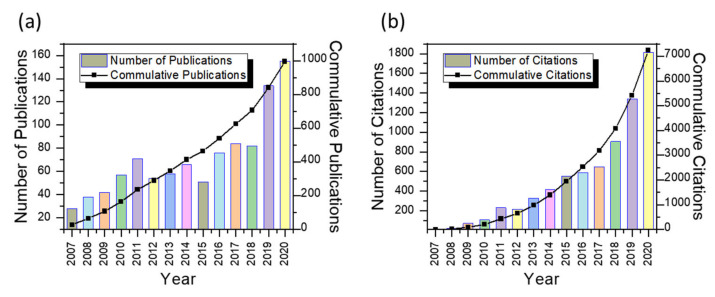
Annual publications and citation output: (**a**) number of publications and their cumulative from 2007 to 2020 and (**b**) number of citations and their cumulative from 2007 to 2020.

**Figure 2 materials-15-02910-f002:**
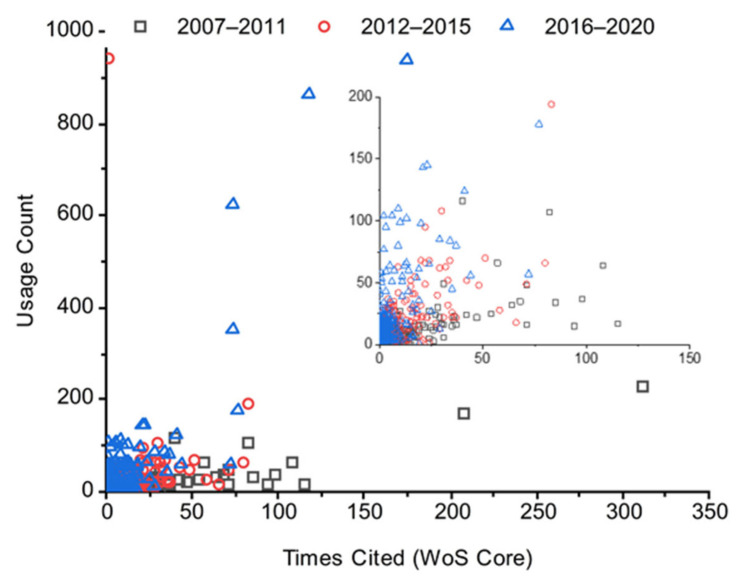
Usage count versus time cited for three different periods, i.e., 2007–2011 (old papers), 2012–2015 (recent papers), and 2016–2020 (current papers).

**Figure 3 materials-15-02910-f003:**
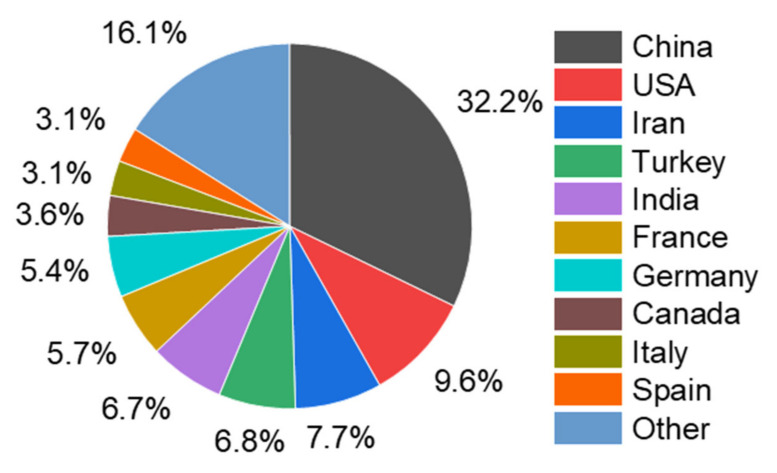
Most influential countries in terms of publications.

**Figure 4 materials-15-02910-f004:**
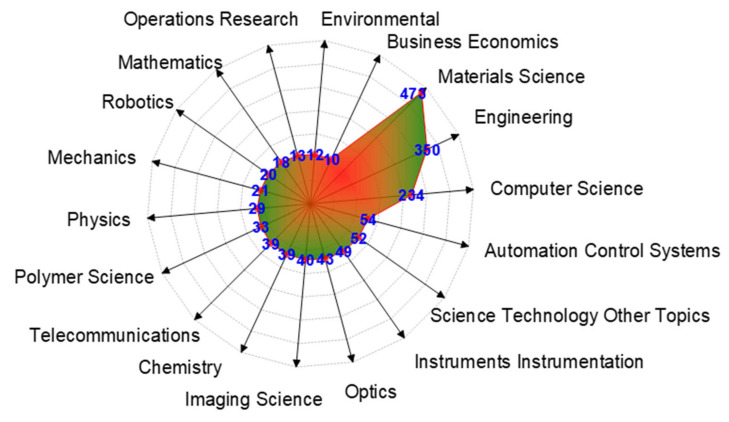
Subject categories according to Research areas in the Web of Science.

**Figure 5 materials-15-02910-f005:**
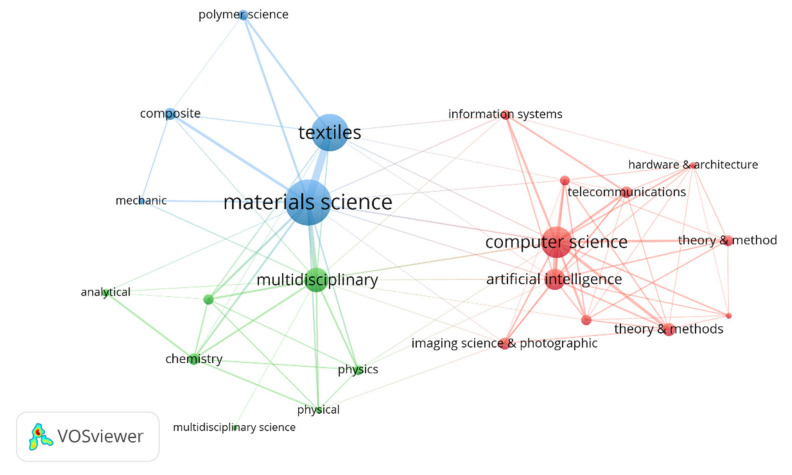
Co-occurrence map based on the Web of Science category.

**Figure 6 materials-15-02910-f006:**
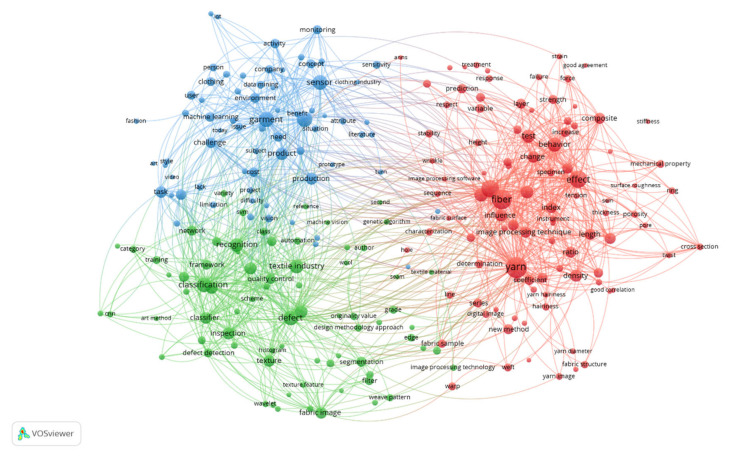
Co-occurrence analysis of research themes from the abstracts of the dataset.

**Table 1 materials-15-02910-t001:** Document types of final manuscripts.

Document Type	Article	Proceedings Paper	Early Access	Review
Records	649	343	24	21
Rate %	65.161	34.438	2.41	2.108

**Table 2 materials-15-02910-t002:** Highly cited research papers.

No.	Title	Journal	Year	Time Cited		Usage Count	Reference
				WoS Core	WoS	Since 2013	
1	Computer-vision-based fabric defect detection: A survey	*IEEE Trans. Ind. Electron.*	2008	312	356	230	[45]
2	Automated fabric defect detection-A review	*Image Vis. Comput.*	2011	208	240	168	[46]
3	Stretchable Ti3C2Tx MXene/Carbon Nanotube Composite Based Strain Sensor with Ultrahigh Sensitivity and Tunable Sensing Range	*ACS Nano*	2018	175	177	934	[47]
4	Fiber/Fabric-Based Piezoelectric and Triboelectric Nanogenerators for Flexible/Stretchable and Wearable Electronics and Artificial Intelligence	*Adv. Mater.*	2020	118	119	864	[48]
5	Exploiting Data Topology in Visualization and Clustering Self-Organizing Maps	*IEEE Trans. Ind. Electron.*	2009	115	116	17	[49]
6	Autonomic healing of low-velocity impact damage in fiber-reinforced composites	*Compos. Part-A Appl. Sci. Manuf.*	2010	108	109	64	[50]
7	Majority Voting: Material Classification by Tactile Sensing Using Surface Texture	*IEEE Trans. Robot.*	2011	98	100	37	[51]

**Table 3 materials-15-02910-t003:** Top 15 institutes publishing in this research domain.

No.	Name of University	Number of Publications	Rate (%)
1	Donghua University	51	5.115
2	Jiangnan University	41	4.112
3	Isfahan University Technology	32	3.21
4	Soochow University	25	2.508
5	Hong Kong Polytech University	24	2.407
6	Shanghai University Engineering and Science	22	2.207
7	Amirkabir University Technology	21	2.106
8	University Lille Nord France	13	1.304
9	ENSAIT	12	1.204
10	RWTH Aachen	12	1.204
11	University of Minho	12	1.204
12	Indian Institute of Technology	11	1.103
13	Technical University of Liberec	11	1.103
14	Tiangong University	11	1.103
15	Xian Polytech University	11	1.103
16	Other	687	69.007

**Table 4 materials-15-02910-t004:** The top 8 journals publishing in this research domain.

No.	Name of Journal	Number of Publications	Impact Factor	Rate (%)
1	*Journal of the Textile Institute*	73	1.239	7.322
2	*Textile Research Journal*	59	1.66	5.918
3	*Fibres Textiles in Eastern Europe*	29	0.76	2.909
4	*Fibers and Polymers*	23	1.59	2.307
5	*Advanced Materials Research*	21	--	2.106
6	*Proceedings of SPIE*	20	0.56	2.006
7	*International Journal of Clothing Science and Technology*	19	0.92	1.906
8	*Indian Journal of Fibre Textile Research*	18	0.6	1.805

## Data Availability

The corresponding author can provide the data on request.

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
