# Peer review of "Bibliometric Analysis of Artificial Intelligence in Textiles"

_materials, 2022, doi:10.3390/ma15082910_

Round 1

Reviewer 1 Report

The article overviews artificial intelligence in the field of textiles over the last 14 years and shall remain for particular interest in the upcoming year. The study is relevant, because the review of articles on the presented topic was performed. However, I have a few recommendations to the authors:

  1. Even 996 articles were found on the presented topic, but only 40 references are cited.
  2. What means the second (smaller) graph in Figure 2?
  3. It would be interesting to know, what topics were analysed in which countries and in which institutions.
  4. I recommend to extend the list of References and to use them in subsections 3.4.1, 3.4.2 and 3.4.3.

Author Response

Comments 1

The article overviews artificial intelligence in the field of textiles over the last 14 years and shall remain for particular interest in the upcoming year. The study is relevant, because the review of articles on the presented topic was performed. However, I have a few recommendations to the authors:

Response to the reviewer

We are thankful to the reviewer for devoting valuable time to give valuable insights.

Comments 2

Even 996 articles were found on the presented topic, but only 40 references are cited.

Response to the reviewer

We have enriched the reference section.

Comments 3

What means the second (smaller) graph in Figure 2?

Response to the reviewer

The second (smaller) graph is the same as the original one; we just zoomed it in so that readers might see it in detail as the bigger graph show the data in the larger context.

Comments 4

It would be interesting to know, what topics were analysed in which countries and in which institutions.

Response to the reviewer

Yes, we agree with the reviewer’s idea that a complete study can be made to analyze and understand the topic country-wise; we will consider this valuable advice in our upcoming research.

Comments 5

I recommend to extend the list of References and to use them in subsections 3.4.1, 3.4.2 and 3.4.3.

Response to the reviewer

The section has been extended according to the suggestion of the reviewer. We hope the reviewer will find it acceptable in its present form.

Reviewer 2 Report

The article is well written.
I suggest you place the reference near the citation (Line 56 on page 2) and make Table 2 more understandable.
If the article presented the main findings among the evaluated works, it would be more interesting.

Author Response

Comments 1

The article is well written.

Response to the reviewer

We are thankful to the reviewer for his encouraging comment.

Comments 2

I suggest you place the reference near the citation (Line 56 on page 2) and make Table 2 more understandable.

Response to the reviewer

We have added citations and explained Table 2 in more detail.

Comments 3

If the article presented the main findings among the evaluated works, it would be more interesting.

Response to the reviewer

Yes, the article presented the main findings, and we submitted them as highlights. We have removed section “highlights” based on the suggestion of reviewer 3 and moved it to a conclusion, as per the guidance of Reviewer 2.

Reviewer 3 Report

  1. Habiba and co-workers’ manuscript entitled, a bibliometric analysis of artificial intelligence in textiles, has been read carefully. I believe the article is of great significance and practical worth in textile academia and artificial intelligence.
  2. The work will offer new references in electronics, material science, and textile; meanwhile, it will be beneficial to choose future research directions for researchers. However, there are some corrections needed before its publication.
  3. The section “highlights” should be deleted, as MDPI-Materials do not request it.
  4. The keyword image processing should be replaced by artificial intelligence as image processing is only a small part of artificial intelligence.
  5. Consider adding color to Figure 1 to make it attractive appealing.
  6. Please redescribe Figure 1a. From Figure 1a, I deem the number of publications maintain stable from 2016 to 2018. The number is increased dramatically from 2018 to 2020.
  7. Consider preparing the article according to the format of MDPI- Materials.
  8. All the tables should maintain a uniform standard. For example, there is a remarkable difference between Tables 1 and 2. There is an unnecessarily thick line in the header of Table 2.
  9. Consider enriching reference section, not necessarily manuscripts found on the web of science, but also other related non-SCI manuscripts.
  10. Please check all references for formatting. For example, ref. 10 and 13 have no publication date. Ref. 12 does not have the number of pages.

Author Response

Comments 1

Habiba and co-workers’ manuscript entitled, a bibliometric analysis of artificial intelligence in textiles, has been read carefully. I believe the article is of great significance and practical worth in textile academia and artificial intelligence.

Response to the reviewer

We are thankful to the reviewer for encouraging comments.

Comments 2

The work will offer new references in electronics, material science, and textile; meanwhile, it will be beneficial to choose future research directions for researchers. However, there are some corrections needed before its publication.

Response to the reviewer

We have revised the manuscript according to the guidance of anonymous reviewers’ guidelines. We believe the reviewers will find the article considerable in revised form.

Comments 3

The section “highlights” should be deleted, as MDPI-Materials do not request it.

Response to the reviewer

We have removed section “highlights” based on the suggestion of reviewer 3 and moved it to a conclusion, as per the guidance of Reviewer 2.

Comments 4

The keyword image processing should be replaced by artificial intelligence as image processing is only a small part of artificial intelligence.

Response to the reviewer

We appreciate the valuable suggestion of the reviewer and have replaced the keyword accordingly.

Comments 5

Consider adding color to Figure 1 to make it attractive appealing.

Response to the reviewer

We have tried to make it more attractive/appealing. Please find colored Figure in revised manuscript.

Comments 6

Please redescribe Figure 1a. From Figure 1a, I deem the number of publications maintain stable from 2016 to 2018. The number is increased dramatically from 2018 to 2020.

Response to the reviewer

This is what the data was obtained from WoS as per query search in Appendix, and it is beyond our control.

Comments 7

Consider preparing the article according to the format of MDPI- Materials.

Response to the reviewer

Sure, we will do that.

Comments 8

All the tables should maintain a uniform standard. For example, there is a remarkable difference between Tables 1 and 2. There is an unnecessarily thick line in the header of Table 2.

Response to the reviewer

Thanks, this will be corrected while revising the format.

Comments 9

Consider enriching reference section, not necessarily manuscripts found on the web of science, but also other related non-SCI manuscripts.

Response to the reviewer

We agree; we have enriched the reference section.

Comments 10

Please check all references for formatting. For example, ref. 10 and 13 have no publication date. Ref. 12 does not have the number of pages.

Response to the reviewer

We have carefully double-checked the completeness of all the references.

Round 2

Reviewer 1 Report

The article was improved. Remarks were taken into account.

Author Response

We are thankful for the reviewer for giving encouraging comments.